# Synthesis of DPIE [2-(1,2-Diphenyl-1*H*-indol-3-yl)ethanamine] Derivatives and Their Regulatory Effects on Pro-Inflammatory Cytokine Production in IL-1β-Stimulated Primary Human Oral Cells

**DOI:** 10.3390/molecules27030899

**Published:** 2022-01-28

**Authors:** Jeongah Lim, Jihyoun Seong, Seunggon Jung, Tae-Hoon Lee, Eunae Kim, Sunwoo Lee

**Affiliations:** 1Department of Chemistry, Chonnam National University, Gwangju 61186, Korea; 91jeongah@gmail.com; 2Department of Oral Biochemistry, Dental Science Research Institute, School of Dentistry, Chonnam National University, Gwangju 61186, Korea; luvhhy@hanmail.net (J.S.); thlee83@jnu.ac.kr (T.-H.L.); 3Department of Oral & Maxillofacial Surgery, School of Dentistry, Chonnam National University, Gwangju 61186, Korea; seunggon.jung@jnu.ac.kr; 4College of Pharmacy, Chosun University, Gwangju 61452, Korea

**Keywords:** IL-1, IL-1R1, pro-inflammatory cytokine, DPIE derivative

## Abstract

Interleukin-1 beta (IL-1β) has diverse physiological functions and plays important roles in health and disease. In this report, we focus on its function in the production of pro-inflammatory cytokines, including IL-6 and IL-8, which are implicated in several autoimmune diseases and host defense against infection. IL-1β activity is markedly dependent on the binding affinity toward IL-1 receptors (IL-1Rs). Several studies have been conducted to identify suitable small molecules that can modulate the interactions between 1L-1β and 1L-1R1. Based on our previous report, where DPIE [2-(1,2-Diphenyl-1*H*-indol-3-yl)ethanamine] exhibited such modulatory activity, three types of DPIE derivatives were synthesized by introducing various substituents at the 1, 2, and 3 positions of the indole group in DPIE. To predict a possible binding pose in complex with IL-1R1, a docking simulation was performed. The effect of the chemicals was determined in human gingival fibroblasts (GFs) following IL-1β induction. The DPIE derivatives affected different aspects of cytokine production. Further, a group of the derivatives enabled synergistic pro-inflammatory cytokine production, while another group caused diminished cytokine production compared to DPIE stimulation. Some groups displayed no significant difference after stimulation. These findings indicate that the modification of the indole site could modulate IL-1β:IL1R1 binding affinity to reduce or enhance pro-inflammatory cytokine production.

## 1. Introduction

Bacterial infection in oral tissues causes periodontal diseases and may lead to chronic inflammation or alveolar bone loss [1]. Therefore, preventing bacterial infection or regulating inflammation is necessary. Oral bacteria, including *Fusobacterium nucleatum* and *Porphyromonas gingivalis*, form a crucial bacterial complex in the pathogenesis of periodontal disease [2,3]. The bacteria complex induces gingival inflammation by secreting virulence factors, mainly LPS, which mediate the secretion of pro-inflammatory cytokines from oral fibroblasts and immune cells [4]. The oral cells release interleukin-1 beta (IL-1β), which triggers IL-1R1 to produce chemokines as well as IL-1β itself in response to LPS stimulation [5]. IL-1β binds to IL-1R1, forming a heterodimer with IL-1 type 3 receptor (IL-1R3). The heterodimer initiates intracellular signaling via Myd88, finally inducing pro-inflammatory cytokine gene expression [6,7]. IL-6 and IL-8 are the prominent cytokines closely related to tissue damage and alveolar bone loss [8,9].

Periodontitis is currently treated by removing the pathogenic biofilm; however, this strategy is not effective. Therefore, many studies have sought to develop new therapeutics to control infection and inflammation. In fact, several trials have been conducted to control inflammatory responses against IL-1β:IL-1R interactions. The therapies target IL-1 receptors, IL-1β, or upstream of the IL-1β signal. For instance, anakinra, AMG 108, and EBI-005 bind to IL-1R1 and inhibit its interaction with IL-1β [10,11,12,13]. Rilonacept, canakinumab, gevokizumab, and LY2189102 target IL-1β, and as they are monoclonal antibodies, they can neutralize IL-1β [14,15,16]. VX-765 targets caspase-1, which directly activates IL-1β [17]. Despite their high ligand specificity, most drugs are proteins, including therapeutic antibodies. There are two major limitations of protein drugs: short serum half-life and low activity. Accordingly, a high dose is often required, which might induce an immune response against the protein drugs. The low productivity of protein drugs leads to high medical expenses. To resolve these limitations, small organic molecules have received considerable attention as potential drugs because structural modification is possible with these molecules. Accordingly, in our study, we opted to focus on synthetic drugs for regulating IL-1β:IL-1R signals.

In our previous study, 60 molecules were evaluated to determine the effect of pro-inflammatory cytokine production in IL-1β-stimulated gingival fibroblasts (GFs) using in silico computational analysis [18]. Among them, only DPIE had a positive effect on pro-inflammatory cytokine expression in IL-1β-stimulated GFs. DPIE increased IL-6 and IL-8 production by 3-fold and 2-fold, respectively, compared to the control. Owing to these results, we opted to determine the correlation between the chemical structure of DPIE and its binding affinity to receptors. The chemical structure of DPIE consists of an indole core in which phenyl groups are attached to N and 2-positions, and the 3-aminoethyl group is connected to the 3-position, as shown in Figure 1. In the prediction of DPIE, which can regulate IL-1β:IL-1R interaction, DPIE interacts with Glu 129 in the immunoglobulin-like domains (D2) of IL-1R via hydrogen bonds and the Asp128 residue in IL-1β. Therefore, we sought to prepare three types of DPIE derivatives: Type 1 is a DPIE derivative with a different N-aryl substituent; Type 2 has two different substituents at the 3-position of the indole core of DPIE, an amine group (type 2-1), and an alkyl group (type 2-2); and Type 3 is the derivative that does not have a phenyl substituent at the 2-position of the indole core.

## 2. Results and Discussion

### 2.1. Synthesis of DPIE Derivatives

To synthesize Type 1 derivatives, we attempted to use the 2-phenyl-1*H*-indole derivative with the 2-aminoethyl group at the 3-position of indole as an intermediate and induce a reaction with aryl halides. However, the C-N bond formation of aryl halides and the intermediate did not result in the desired product. Therefore, C-N bond formation between aryl halides and 2-phenyl-1*H*-indole was conducted, and then a 2-aminoethyl group was introduced at the 3-position of indole via three steps. The results are summarized in Figure 1. We confirmed that DPIE could be obtained by using these reaction processes. To prepare Type 1 compounds, three types of aryl iodides, including 4-iodo-1,1-biphenyl, 4-trifluromethylphenyl iodide, and 2-iodo-9,9-dimethyl-9*H*-fluorene, were selected and allowed to react with **1** in the presence of a CuI catalyst. The desired coupled products **2b**, **2c**, and **2d** were obtained, with yields of 90%, 79%, and 81%, respectively. Thereafter, formylation, condensation with nitromethane, and reduction were conducted, and the desired Type 1 products, including **4b**, **4c**, and **4d**, were successfully obtained. 

Type 2-1 compounds with N-substituents at the amino ethyl group were synthesized from 2-phenyl-1*H*-indole via a three-step reaction process (Figure 2). Copper-catalyzed *N*-phenylation of **1** resulted in **2a** with a yield of 77%. Further, formylation of **2a** resulted in 3-formylated indole derivate **3**, with a yield of 70%. The reduction aminations with primary or secondary amines produced the desired Type 2-1 compounds in good yields. 

Type 2-2 and Type 3 compounds were prepared using the Fisher indole synthetic method (Figure 3). Phenyl hydrazine was reacted with ketones or aldehydes. When alkyl phenyl ketones were allowed to react with phenyl hydrazine, 3-alkyl-substituted 2-phenyl-1*H*-indole derivatives were formed in good yields. Alkyl aldehydes, such as butyraldehyde, pentanal, and 4-aminobutanal, afforded the 3-alkyl-substituted-1*H*-indoles in good yields. Finally, these 3-alkyl-substituted 1*H*-indoles were treated with phenyl iodide in the presence of copper-catalyst to yield the corresponding *N*-phenyl indoles, **9** and **12**. 

### 2.2. Effect of DPIE Derivatives on Pro-Inflammatory Cytokine Production in IL-1β-Stimulated GFs

After synthetic processes, the derivatives were screened using human GFs, which are the most abundant structural cells in the periodontal tissue and cause persistent inflammation in periodontal disease, to discover new therapeutics that regulate IL-1β:IL-1R1 interaction. To determine the activity of small molecules, GFs were pre-incubated with the derivatives at the indicated concentration. The concentration of small molecules was determined after cell cytotoxic assay (Appendix A), the molecules were applied with non-cytotoxic concentrations. Then cells were stimulated with IL-1β for 8 h. Cells were harvested, and the mRNA expression of pro-inflammatory cytokines (IL-6 and IL-8) was determined.

Type 1 modification tended to increase IL-6 mRNA but had no effect on IL-8 mRNA expression. As shown in Figure 2, **4c** and **4d** in Type 1 derivatives had better activity in IL-6 production than DPIE. However, only compound **4d** induced a larger production of IL-8 than DPIE. These results suggest that the binding affinity between IL-1β and IL-1R might be increased as the size of the substituent at the *N*-position increased. The findings indicate that Type 1-**4d** modification, which involves increasing the size of the substituent at the *N*-position, could provide a better structure for developing an IL-1β: IL-1R1 interaction enhancer.

Type 2-1 compounds were expected to exhibit hydrophilic and hydrogen bonding affinity between amino substituents and target proteins, while Type 2-2 compounds were expected to exhibit a hydrophobic interaction between the alkyl group substituents and target protein. As shown in Figure 3, when the Type 2-1 and Type 2-2 compounds were evaluated in the IL-1β-induced inflamed GFs, all compounds exhibited lower activity than DPIE in the production of both IL-6 and IL-8. When these compounds were administered, the mRNA levels of cytokines were similar to those obtained by IL-1β stimulation alone. Such findings indicate that Type 2 compounds could not contribute to the receptor:ligand interaction. 

As shown in Figure 4, Type 3 compounds, which do not have a phenyl ring at the 2-position, induced a greater decrease in the expression of IL-6 and IL-8 than IL-1β alone. The statistical significance, the *p*-value, between IL-1β alone and Type 3 compounds less than 0.05 in IL-6 mRNA expression indicated as asterisk. The IL-8 mRNA expression also showed decreasing tendency after Type 3 compound treatment; **12c** had a great effect on suppressing both IL-6 and IL-8 mRNA expression, the *p*-value of **12c** compared to IL-β alone was less than 0.05, indicated as asterisk. These results support that the phenyl ring at the 2-position of indole moiety was a key functionality to increase the binding affinity between DPIE and receptors. This modification could be a good framework for anti-inflammatory agents.

### 2.3. Molecular Docking Study of DPIE Derivatives and Human IL-1R

To predict a binding interaction of DPIE derivatives such as **4d** of Type 1, **5a** of Type 2-1, and **9a** of Type 2-2 with human IL-1R1, the docking study was performed using Vinardo software [19]. According to our previous study [18], the significant residue of the binding sites in IL-1R1 were defined by Pro26 of Site 1 (S1) and Tyr127 of Site 2 (S2). The regions of S1 and S2 were located at the binding interface of IL-1R1 and IL-1β. Regarding the binding pocket S1, it consisted of the inside hydrophobic hole, and the cave was restricted. Meanwhile, the region of S2 coincided with the weak interface in the binding of the IL-1R1 and IL-1β and made up of hydrophobic cores. After running docking calculations for two binding sites, the best pose was selected based on their docking score and analyzed concerning the derailed information about the binding mode in the cavity. Depending on the score value, the best poses of the derivatives **5a** and **9a,** such as Type 2-1 and Type 2-2, were occupied in S1, while **4d** of Type 1 was bound to S2 (Figure 5). The binding affinity was −6.9 kcal/mol, −7.2 kcal/mol, and −9.3 kcal/mol for **4d**, **5a**, and **9a**, respectively. Regarding the binding pose of two different Type 2 compounds in S1, 2-phenyl-*N*-phenyl indole moiety of **5a** and **9a** was surrounded by the hydrophobic sidechains of IL-1R1, such as Ile13, Leu15, Pro26, Pro28, Tyr127, and Phe130, in common. The difference of the binding pose between **5a** and **9a** was that the amine group of **5a** (Type 2-1) slightly formed a hydrogen bond with the charged side chain of Glu129, and the alkyl group of **9a** was between Glu129 and Phe130. Further, since two derivatives were deeply buried in the S1 of IL-1R1, it could not affect the IL-1R1 and IL-1β interaction directly. Compared with **5a** and **9a**, interestingly, the best pose of **4d** was tightly stuck to S2. The indole group of **4d** lay on the hydrophobic core, including Pro26, Tyr127, and Phe130, and the 9,9-dimethylfluoren moiety was reached on Ile13 and Leu15. In addition, the ethanamine group of **4d** formed a hydrogen bond with the backbone of Lys112, and the 2-phenyl group was stacked with the alkyl group of Lys112 in parallel. It is an important binding region formed of the hydrophobic cores, and the derivatives **4d** could construct in the stable complex with IL-1R1 because S2 seems like a crack between IL-1R1 and IL-1β. Noticeably, the possible binding poses of **4d** only occurred in S2, and the binding affinity was lower than that of **5a** and **9a**. Therefore, the analog expansion of **4d** could be promised to enhance the protein-protein interaction of IL-1R1 and IL-1β.

## 3. Materials and Methods

### 3.1. Ethics Statement

Chonnam National University Dental Hospital Institutional Review Board (Approval No., CNUDH-2016-014; 21 November 2016) approved the isolation of human GFs. Written informed consent was obtained from all individuals after the nature, and possible consequences of the studies were explained. 

### 3.2. Cell Culture and Reagents

Primary human GFs were cultured in Dulbecco’s Modified Eagle’s Medium (DMEM, Gibco BRL, Grand Island, NY, USA) supplemented with 10% Fetal bovine serum (heat-inactivated, PAA Laboratories, Etobicoke, ON, Canada), 100 U/mL Penicillin, and 100 μg/mL Streptomycin (Gibco BRL) at 37 °C in a humidified 5% CO_2_. The cells were pre-incubated with DPIE derivatives and stimulated with 0.5 ng/mL human IL-1β (Peprotech, Korea) for 8 h. The cells were then washed with phosphate-buffered saline and lysed using TRIzol reagent (Invitrogen, Calsbad, CA, USA) for mRNA purification.

### 3.3. Reverse-Transcription PCR

Total RNA was isolated using TRIzol reagent. A total of 500 ng RNA was reverse-transcribed into cDNA. Briefly, RNA, oligo dT (Takara, Japan) primer, and dNTP mixture were stored at 65 °C for 5 min and then cooled immediately on ice. Thereafter, PrimeScript RTase (Takara, Japan) was added, and the mixture was incubated under the following conditions: 30 °C for 10 min, 42 °C for 1 h, and 95 °C for 5 min. To determine IL-6 and IL-8 mRNA expression levels, reverse-transcription PCR was performed with the following primers: IL-6, 5′-AGGGCTCTTCGGGAAATGTA-3′, and 5′-TGCCCAGTGGACAGGTTTC-3′; and IL-8, 5′-CCATAAGGCACAAACTTTCA-3′, and 5′-GTTCCTTCCGGTGGTTTCTTC-3′. GAPDH was used as a reference for quantitative analysis. PCR was conducted using the QuantStudio 3 qRT-PCR system (Applied Biosystems, Foster City, CA, USA), the Power SYBR Green PCR Master Mix (4367659; Applied Biosystems, Foster City, CA, USA), and a temperature protocol provided by the company. The cycle threshold values are expressed as relative ratios and were calculated using the 2^−ΔΔCt^ method.

### 3.4. Statistical Analysis

Data were represented as means of ±standard deviation (SD). Statistical significance was determined by student’s *t*-test, and *p*-values of less than 0.05 were considered statistically significant. The significance values are indicated by asterisks in figures (* *p* < 0.05; ** *p* < 0.01). All experiments were repeated in at least 3 independent experiments.

### 3.5. Synthetic Method

#### 3.5.1. General Information

All reagents were purchased and used without further purification. Analytical thin-layer chromatography (TLC) was performed on silica gel (Merck Kieselgel 60 F254) pre-coated aluminum plates, and the products were visualized by short-wave (254 nm) or long-wave (360 nm) UV light. Flash column chromatography was performed using silica gel (Merck, Kieselgel 60, 230–240 mesh). ^1^H spectra were recorded in CDCl_3_ or DMSO-d_6_ on 500 MHz NMR spectrometers, and data are reported as follows: chemical shift, multiplicity (s = singlet, d = doublet, t = triplet, m = multiplet), coupling constants (Hz), and integration. ^13^C spectra were recorded in CDCl_3_ on 126 MHz NMR spectrometers, and resonances (δ) are given in ppm. ^1^HNMR and ^13^CNMR were recorded on Agilent Technologics (Santa Clara, CA, USA) NMR spectrometer (500 MHz for ^1^H). High-resolution mass spectra were recorded on a time of flight (TOF) mass spectrometer (JEOL, Model: JMS-T200GC). The purities of all DPIE derivatives were determined by HPLC (Waters ACQUITY (UPLC), SQD2) and confirmed to 100%.

#### 3.5.2. General Procedure I: Synthesis of DPIE Derivative Type 1

2-Phenyl indole (1.0 mmol, 1.0 equiv), aryl iodide (1.0 mmol, 1.0 equiv), CuI (19.0 mg, 0.1 mmol, 10 mol%), trans-1,2-cyclohexane diamine (34 mg, 0.3 mmol, 30 mol%), and K_3_PO_4_ (425 mg, 2.0 mmol, 2.0 equiv) were reacted in toluene (10.0 mL) at 120 °C for 12 h. The reaction solution was quantitatively transferred to a separatory funnel with EtOAc and water. The layers were mixed and allowed to separate. Thereafter, the organic layer was washed three times with water, dried with MgSO_4_, filtered, and concentrated in vacuo to yield the crude product, which was purified by column chromatography on silica gel (eluent: hexane/ethyl acetate = 4/1) to yield the *N*-arylated indole **2**. Compound **2** (1.0 mmol) was reacted with POCl_3_ (184 mg, 1.2 mmol, 1.2 equiv) in DMF (10.0 mL) at 25 °C for 2 h. The reaction mixture was treated with water and 6 N NaOH until the pH was 11. The resulting solid was collected by filtration, washed with cold H_2_O, and dried under vacuum to yield the formulated compound **3**. Compound **3** (1.0 mmol, 1 equiv) and NH_4_OAc (1.2 mmol, 1.2 equiv) were reacted in CH_3_NO_2_ (10.0 mL) at 100 °C for 1 h. The resulting solid was collected by filtration, washed with H_2_O, and dried under vacuum to yield the nitro vinyl compounds. The resulting orange powder (1.0 mmol, 1.0 equiv) was subsequently dissolved in THF (10 mL) and cooled to 0 °C. LiAlH_4_ (4 mmol, 4.0 equiv) was added portion-wise, and after complete addition, the reaction was heated to reflux at 60 °C for 6 h. The reaction mixture was cooled to 0 °C, quenched by the careful addition of 15% NaOH (5 mL) and water (3 mL), and diluted with EtOAc. The organic layer was washed three times with water, dried with MgSO_4_, filtered, concentrated in vacuo, and the residue was purified by flash column chromatography on silica gel (eluent: MC/MeOH, with 10% NH_4_OH). The residue was then dried with MgSO_4_, filtered, and concentrated in vacuo, yielding compound **4**.

##### 2-(1,2-Diphenyl-1*H*-indol-3-yl)ethanamine (**4a**: DPIE)

General procedure I with iodobenzene (204 mg, 1.0 mmol) afforded 2-(1,2-Diphenyl-1*H*-indol-3-yl)ethanamine (DPIE) (47 mg, 0.15 mmol, 15% yield) as a yellow oil; ^1^H NMR (500 MHz, CDCl_3_) δ 7.78–7.75 (m, 1H), 7.39–7.34 (m, 3H), 7.33–7.26 (m, 6H), 7.26–7.19 (m, 4H), 3.12–2.96 (m, 4H), 1.34 (s, 2H); ^13^C NMR (126 MHz, CDCl_3_) δ 138.4, 138.2, 137.7, 132.1, 130.8, 129.1, 128.4, 128.2, 128.0, 127.6, 126.8, 122.6, 120.3, 119.2, 112.7, 110.7, 43.2, 29.3; HRMS (FD-TOF) *m*/*z*[M]^+^ calcd for C_22_H_20_N_2_: 312.1626, found: 312.1624.

##### 2-(1,2-Diphenyl-1*H*-indol-3-yl)ethanamine (**4b**)

General procedure I with 4-iodo-1,1′-biphenyl (280 mg, 1.0 mmol) afforded 2-(1,2-Diphenyl-1*H*-indol-3-yl)ethanamine (**4b**)) (242 mg, 0.7 mmol, 70% yield) as a yellow oil; ^1^H NMR (500 MHz, CDCl_3_) δ 7.74–7.71 (m, 1H), 7.57–7.49 (m, 4H), 7.42–7.36 (m, 3H), 7.33–7.29 (m, 1H), 7.28–7.25 (m, 4H), 7.25–7.17 (m, 5H), 3.06–2.95 (m, 4H), 1.44 (s, 2H); ^13^C NMR (126 MHz, CDCl_3_) δ 140.1, 139.4, 138.1, 137.7, 137.6, 132.1, 130.8, 128.9, 128.5, 128.3, 128.1, 127.7, 127.6, 127.1, 122.8, 120.4, 119.3, 112.9, 110.8, 43.1, 29.2; HRMS (FD-TOF) *m/z*[M]^+^ calcd for C_28_H_24_N_2_: 388.1939, found: 388.1938.

##### 2-(2-phenyl-1-(4-(trifluoromethyl)phenyl)-1*H*-indol-3-yl)ethanamine (**4c**)

General procedure I with 4-iodobenzotrifluoride (272 mg, 1.0 mmol) afforded 2-(2-phenyl-1-(4-(trifluoromethyl)phenyl)-1*H*-indol-3-yl)ethanamine (**4c**)) (186 mg, 0.6 mmol, 55% yield) as a yellow oil; ^1^H NMR (500 MHz, CDCl_3_) δ 7.78–7.74 (m, 1H), 7.61 (d, *J* = 8.3 Hz, 2H), 7.40–7.29 (m, 6H), 7.28–7.23 (m, 4H), 3.10–2.97 (m, 4H), 1.41 (s, 2H);^13^C NMR (126 MHz, CDCl_3_) δ 141.7, 137.7, 137.4, 131.6, 130.7, 128.7, 128.5, 128.5 (q, *J* = 32.9 Hz), 128.0, 127.9, 126.3 (q, *J* = 3.7 Hz), 124.0 (q, *J* = 272.2 Hz), 123.1, 120.9, 119.5, 114.0, 110.4, 43.1, 29.2; HRMS (FD-TOF) *m*/*z*[M]^+^ calcd for C_23_H_19_F_3_N_2_: 380.1500, found: 380.1495.

##### 2-(1-(9,9-dimethyl-9*H*-fluoren-2-yl)-2-phenyl-1*H*-indol-3-yl)ethanamine (**4d**)

General procedure I with 2-bromo-9,9-dimethyl-9*H*-fluorene (273 mg, 1.0 mmol) afforded 2-(1-(9,9-dimethyl-9*H*-fluoren-2-yl)-2-phenyl-1H-indol-3-yl)ethanamine (**4d**) (181 mg, 0.5 mmol, 47% yield) as a yellow oil; ^1^H NMR (500 MHz, CDCl_3_) δ 9.98 (s, 2H), 8.54 (dt, *J* = 7.8, 1.0 Hz, 2H), 7.79–7.70 (m, 4H), 7.45–7.28 (m, 23H), 7.07 (dd, *J* = 1.9, 0.5 Hz, 2H), 1.31 (s, 12H); ^13^C NMR (126 MHz, CDCl_3_) δ 154.5, 153.8, 138.5, 138.2, 137.5, 137.2, 132.2, 130.8, 128.4, 128.1, 127.5, 127.4, 127.1, 126.2, 122.8, 122.6, 120.3, 120.2, 120.1, 119.2, 112.4, 110.7, 46.8, 43.0, 28.9, 26.8. MS (ESI) *m*/*z*: 428.2[M]^+^.

#### 3.5.3. General Procedure II: Synthesis of DPIE Derivative Type 2-1

Compound **3** (1.0 mmol, 1.0 equiv), amine, and AcOH (1.5 mmol, 1.5 equiv) were reacted in DCE at 25 °C for 0.5 h. Thereafter, NaHB(OAc)_3_ (1.5 mmol, 1.5 equiv) was added to the reaction mixture. The resulting mixture was treated with aqueous Na_2_CO_3_ and EtOAc. The organic layer was separated and washed three times with water, dried with MgSO_4_, filtered, and concentrated in vacuo to yield the crude product, which was purified by column chromatography on silica gel (eluent: hexane/ethyl acetate = 4/1 v/v) to afford compound **5**.

##### *N*-((1,2-Diphenyl-1*H*-indol-3-yl)methyl)cyclopropanamine (**5a**)

General procedure II with cyclopropanamine (57 mg, 1.0 mmol) and 1,2-Diphenyl-1*H*-indole-3-carbaldehyde (297 mg 1.0 mmol) afforded *N*-((1,2-Diphenyl-1*H*-indol-3-yl)methyl)cyclopropanamine (**5a**) (115 mg, 0.3 mmol, 34% yield) as a yellow oil; ^1^H NMR (500 MHz, DMSO-D_6_) δ 7.83–7.77 (m, 1H), 7.41 (t, *J* = 7.6 Hz, 2H), 7.37–7.25 (m, 6H), 7.22–7.10 (m, 5H), 3.87 (s, 2H), 2.16–2.08 (m, 1H), 0.40–0.29 (m, 2H), 0.30–0.22 (m, 2H); ^13^C NMR (126 MHz, DMSO-D_6_) δ 138.3, 138.2, 137.7, 131.6, 130.8, 129.8, 128.5, 128.5, 128.3, 128.1, 127.6, 122.9, 120.5, 120.1, 114.3, 110.4, 43.8, 31.4, 30.9, 22.5, 14.5, 6.7; HRMS (FD-TOF) *m*/*z*[M]^+^ calcd for C_24_H_22_N_2_: 338.1783, found: 338.1777.

##### *N*-((1,2-Diphenyl-1*H*-indol-3-yl)methyl)aniline (**5b**)

General procedure II with aniline (93 mg, 1.0 mmol) and 1,2-Diphenyl-1*H*-indole-3-carbaldehyde (297 mg 1.0 mmol,) afforded *N*-((1,2-Diphenyl-1*H*-indol-3-yl)methyl)aniline (**5b**) (161 mg, 0.4 mmol, 43% yield) as a yellow oil; ^1^H NMR (500 MHz, CDCl_3_) δ 7.50 (d, *J* = 7.7 Hz, 1H), 7.37–7.31 (m, 3H), 7.28–7.26 (m, 1H), 7.25–7.15 (m, 8H), 7.13–7.08 (m, 1H), 7.04 (d, *J* = 8.4 Hz, 2H), 6.63–6.58 (m, 2H), 4.10 (s, 2H); ^13^C NMR (126 MHz, CDCl_3_) δ 144.2, 138.7, 138.0, 138.0, 132.0, 132.0, 130.7, 129.2, 129.1, 128.7, 128.1, 128.1, 127.5, 126.8, 122.5, 120.3, 119.9, 115.5, 114.2, 110.5, 30.0; HRMS (FD-TOF) *m*/*z*[M]^+^ calcd for C_27_H_22_N_2_: 374.1783, found: 374.1785.

##### *N*-((1,2-Diphenyl-1*H*-indol-3-yl)methyl)naphthalen-1-amine (**5c**)

General procedure II with naphthalen-1-amine (143 mg, 1.0 mmol) and 1,2-Diphenyl-1*H*-indole-3-carbaldehyde (297 mg 1.0 mmol) afforded *N*-((1,2-Diphenyl-1*H*-indol-3-yl)methyl)naphthalen-1-amine (**5c**) (221 mg, 0.5 mmol, 52% yield) as a yellow oil; ^1^H NMR (500 MHz, CDCl_3_) δ 8.14–8.06 (m, 1H), 7.95–7.88 (m, 1H), 7.53–7.48 (m, 2H), 7.45–7.42 (m, 1H), 7.41–7.35 (m, 3H), 7.32–7.24 (m, 4H), 7.22–7.16 (m, 6H), 7.11 (d, *J* = 7.6 Hz, 1H), 7.10–7.06 (m, 1H), 6.69 (d, *J* = 7.6 Hz, 1H), 4.54 (d, *J* = 0.7 Hz, 2H); ^13^C NMR (126 MHz, CDCl_3_) δ 140.5, 138.6, 138.3, 138.0, 132.7, 131.8, 130.3, 129.1, 128.9, 128.0, 128.0, 127.6, 127.3, 126.7, 126.0, 125.8, 124.6, 124.3, 124.3, 122.5, 121.5, 120.2, 119.9, 112.9, 110.4, 109.8, 27.5; HRMS (FD-TOF) *m*/*z*[M]^+^ calcd for C_31_H_24_N_2_: 424.1939, found: 424.1938.

##### *N*-((1,2-Diphenyl-1H-indol-3-yl)methyl)pyridin-2-amine (**5d**)

General procedure II with pyridin-2-amine (94 mg, 1.0 mmol) and 1,2-Diphenyl-1*H*-indole-3-carbaldehyde (297 mg 1.0 mmol) afforded *N*-((1,2-Diphenyl-1*H*-indol-3-yl)methyl)pyridin-2-amine (5d) (139 mg, 0.4 mmol, 38% yield) as a yellow oil; ^1^H NMR (500 MHz, CDCl_3_) δ 8.09 (d, *J* = 4.6 Hz, 1H), 7.81–7.76 (m, 1H), 7.42–7.26 (m, 7H), 7.26–7.18 (m, 7H), 6.60–6.54 (m, 1H), 6.37 (d, *J* = 8.4 Hz, 1H), 4.67 (s, 2H); ^13^C NMR (126 MHz, CDCl_3_) δ 158.5, 147.5, 139.0, 138.2, 137.9, 137.6, 131.2, 130.7, 129.2, 128.3, 128.1, 128.0, 127.8, 127.2, 123.0, 120.9, 119.4, 112.7, 111.6, 110.8, 107.3, 37.7; HRMS (FD-TOF) *m*/*z*[M]^+^ calcd for C_26_H_21_N_3_: 375.1735, found: 375.1740.

##### *N*-((1,2-Diphenyl-1H-indol-3-yl)methyl)-N-phenylaniline (**5e**)

General procedure VI with Diphenylamine (170 mg, 1.0 mmol) and 1,2-Diphenyl-1*H*-indole-3-carbaldehyde (297 mg 1.0 mmol) afforded *N*-((1,2-Diphenyl-1*H*-indol-3-yl)methyl)-N-phenylaniline (5e) (248 mg, 0.6 mmol, 55% yield) as a yellow oil; ^1^H NMR (500 MHz, CDCl_3_) δ 7.54–7.52 (m, 2H), 7.38–7.32 (m, 7H), 7.29–7.26 (m, 2H), 7.26–7.23 (m, 8H), 7.23–7.20 (m, 8H), 7.20–7.16 (m, 3H), 7.15–7.10 (m, 7H), 6.96–6.93 (m, 4H), 4.14 (s, 4H); ^13^C NMR (126 MHz, CDCl_3_) δ 141.3, 138.5, 137.9, 137.8, 134.0, 131.9, 130.5, 129.1, 129.0, 128.9, 128.0, 127.9, 127.4, 126.7, 122.4, 120.2, 119.7, 117.8, 113.8, 110.4, 30.0; HRMS (FD-TOF) *m*/*z*[M]^+^ calcd for C_33_H_26_N_2_: 450.2096, found: 450.2075.

#### 3.5.4. General Procedure III: Synthesis of DPIE Derivative Type 2-2 and Type 3

Alkyl phenyl ketone **7** or alkyl aldehyde **10** (1.0 mmol, 1.0 equiv), phenyl hydrazine hydrochloride (1.0 mmol, 1.0 equiv), and cyanuric chloride (0.1 mmol, 0.1 equiv) were reacted in ethanol at 80 °C for 4 h. The resulting mixture was treated with water and EtOAc. The organic layer was separated, washed three times with water, dried with MgSO_4_, filtered, and concentrated in vacuo to yield the corresponding compound **8** or **11**. Compound **8** or **11** (1.0 mmol), iodobenzene (204 mg, 1.0 mmol), CuI (19.0 mg, 0.1 mmol, 10 mol%), trans-1,2-cyclohexane diamine (34 mg, 0.3 mmol, 30 mol%), and K_3_PO_4_ (425 mg, 2.0 mmol, 2.0 equiv) were reacted in toluene (10.0 mL) at 120 °C for 12 h. The resulting mixture was treated with water and EtOAc. The organic layer was separated and washed three times with water, dried with MgSO_4_, filtered, and concentrated in vacuo to yield the crude product, which was purified by column chromatography on silica gel (eluent: hexane/ethyl acetate = 4/1) to afford compound **9** or **12**. 

##### 1,2-Diphenyl-3-propyl-1*H*-indole (**9a**)

General procedure III with 1-phenylpentan-1-one (162 mg, 1.0 mmol) afforded 1,2-Diphenyl-3-propyl-1*H*-indole (**9a**) (112 mg, 0.36 mmol, 36% yield) as a yellow oil; ^1^H NMR (500 MHz, CDCl_3_) δ 7.77–7.72 (m, *J* = 3.6 Hz, 1H), 7.39–7.32 (m, *J* = 6.7 Hz, 3H), 7.32–7.26 (m, 4H), 7.26–7.17 (m, 6H), 2.88 (q, *J* = 7.4 Hz, 2H), 1.36 (t, *J* = 7.4 Hz, 3H); ^13^C NMR (126 MHz, CDCl_3_) δ 138.7, 137.8, 136.7, 132.3, 130.7, 129.1, 128.2, 128.1, 128.1, 127.4, 126.7, 122.4, 120.1, 119.3, 117.4, 110.6, 18.1, 16.0; MS (ESI) *m*/*z*: 408.1[M]^+^.

##### 3-butyl-1,2-Diphenyl-1*H*-indole (**9b**)

General procedure III with 1-phenylhexan-1-one (176 mg, 1.0 mmol) afforded 3-butyl-1,2-Diphenyl-1*H*-indole (9b) (123 mg, 0.38 mmol, 38% yield) as a yellow oil; ^1^H NMR (400 MHz, CDCl_3_) δ 7.73–7.66 (m, 1H), 7.34–7.28 (m, 3H), 7.28–7.22 (m, 4H), 7.22–7.14 (m, 6H), 2.86–2.77 (m, 2H), 1.75–1.64 (m, 2H), 1.43–1.31 (m, 2H), 0.88 (t, *J* = 7.3 Hz, 3H); ^13^C NMR (101 MHz, CDCl_3_) δ 138.6, 137.7, 137.0, 132.3, 130.6, 128.9, 128.4, 127.9, 127.2, 126.5, 122.3, 119.9, 119.2, 115.9, 110.5, 33.3, 24.4, 22.9, 13.9; MS (ESI) *m*/*z*: 408.1[M]^+^.

##### 3-hexyl-1,2-Diphenyl-1*H*-indole (**9c**)

General procedure III with 1-phenyloctan-1-one (204 mg, 1.0 mmol) afforded 3-hexyl-1,2-Diphenyl-1*H*-indole (**9c**) (145 mg, 0.41 mmol, 41% yield) as a yellow oil; ^1^H NMR (400 MHz, CDCl_3_) δ 7.73–7.66 (m, 1H), 7.37–7.12 (m, 12H), 2.86–2.76 (m, 2H), 1.77–1.65 (m, 2H), 1.40–1.30 (m, 2H), 1.28–1.21 (m, 4H), 0.85 (t, *J* = 6.9 Hz, 3H);^13^C NMR (101 MHz, CDCl_3_) δ 138.8, 137.8, 137.1, 132.5, 130.8, 129.1, 128.5, 128.1, 127.3, 126.7, 122.4, 120.1, 119.4, 116.1, 110.6, 31.8, 31.2, 29.6, 24.8, 22.8, 14.2; MS (ESI) *m*/*z*: 408.1[M]^+^.

##### 3-octyl-1,2-Diphenyl-1*H*-indole (**9d**)

General procedure III with 1-phenyldecan-1-one (232 mg, 1.0 mmol) afforded 3-octyl-1,2-Diphenyl-1*H*-indole (**9d**) (149 mg, 0.39 mmol, 39% yield) as a yellow oil; ^1^H NMR (500 MHz, CDCl_3_) δ 7.73–7.66 (m, 1H), 7.36–7.10 (m, 12H), 2.84–2.77 (m, 2H), 1.75–1.67 (m, 2H), 1.37–1.31 (m, 2H), 1.29–1.18 (m, 8H), 0.87 (t, *J* = 7.0 Hz, 3H); ^13^C NMR (126 MHz, CDCl_3_) δ 138.7, 137.8, 137.1, 132.4, 130.7, 129.1, 128.5, 128.1, 128.0, 127.3, 126.7, 122.4, 120.1, 119.4, 116.1, 110.6, 32.1, 31.3, 30.0, 29.5, 29.4, 24.8, 22.8, 14.3. HRMS (FD-TOF) *m*/*z*[M]^+^ calcd for C_28_H_31_N: 381.2457, found: 381.2454.

##### 3-decyl-1,2-Diphenyl-1*H*-indole (**9e**)

General procedure III with 1-phenyldodecan-1-one (260 mg, 1.0 mmol) afforded 3-decyl-1,2-Diphenyl-1*H*-indole (**9e**) (143 mg, 0.35 mmol, 35% yield) as a yellow oil; ^1^H NMR (500 MHz, CDCl_3_) δ 7.73–7.66 (m, 1H), 7.31 (dd, *J* = 10.2, 4.7 Hz, 3H), 7.28–7.21 (m, 4H), 7.21–7.12 (m, 6H), 2.80 (t, *J* = 7.4 Hz, 2H), 1.76–1.66 (m, 2H), 1.37–1.31 (m, 2H), 1.30–1.22 (m, 12H), 0.92–0.84 (m, 3H); ^13^C NMR (126 MHz, CDCl_3_) δ 138.7, 137.8, 137.08, 132.4, 130.7, 129.05, 128.5, 128.1, 128.0, 127.3, 126.7, 122.4, 120.1, 119.4, 116.1, 110.6, 32.1, 31.3, 29.9, 29.8, 29.6, 29.5, 24.8, 22.8, 14.3; HRMS (FD-TOF) *m*/*z*[M]^+^ calcd for C_30_H_35_N: 409.2770, found: 409.2766.

##### 3-hexadecyl-1,2-Diphenyl-1*H*-indole (**9f**)

General procedure III with 1-phenyloctadecan-1-one (345 mg, 1.0 mmol) afforded 3-hexadecyl-1,2-Diphenyl-1*H*-indole (**9f**) (183 mg, 0.37 mmol, 37% yield) as a yellow oil; ^1^H NMR (500 MHz, CDCl_3_) δ 7.73–7.66 (m, 1H), 7.35–7.28 (m, 3H), 7.28–7.20 (m, 5H), 7.20–7.14 (m, 5H), 2.84–2.76 (m, 2H), 1.74–1.66 (m, 2H), 1.33–1.24 (m, 24H), 0.88–0.86 (m, 3H); ^13^C NMR (126 MHz, CDCl_3_) δ 138.7, 137.8, 137.09, 132.4, 130.7, 129.1, 128.5, 128.1, 128.0, 127.3, 126.7, 122.4, 120.1, 119.4, 116.1, 110.6, 32.1, 31.8, 31.3, 30.0, 29.9, 29.8, 29.8, 29.8, 29.6, 29.5, 24.8, 22.9, 14.3; HRMS (FD-TOF) *m*/*z*[M]^+^ calcd for C_36_H_47_N: 493.3709, found: 493.3703.

##### 3-ethyl-1-phenyl-1*H*-indole (**12a**)

General procedure III with butyraldehyde (72 mg, 1.0 mmol) afforded 3-ethyl-1-phenyl-1*H*-indole (**12a**) (33 mg, 0.2 mmol, 15% yield) as a yellow oil; ^1^H NMR (400 MHz, CDCl_3_) δ 7.67–7.63 (m, 1H), 7.58–7.53 (m, 1H), 7.50–7.45 (m, 4H), 7.34–7.27 (m, 1H), 7.24–7.12 (m, 3H), 2.84 (qd, *J* = 7.5, 1.0 Hz, 2H), 1.38 (t, *J* = 7.5 Hz, 3H); ^13^C NMR (101 MHz, CDCl_3_) δ 140.2, 136.3, 129.7, 129.1, 126.1, 124.5, 124.2, 122.5, 120.0, 119.9, 119.4, 110.6, 18.4, 14.5; MS (ESI) *m*/*z*: 221.1[M]^+^.

##### 1-phenyl-3-propyl-1*H*-indole(**12b**)

General procedure III with Pentanal (86 mg, 1.0 mmol) afforded 1-phenyl-3-propyl-1*H*-indole (**12b**) (47 mg, 0.2 mmol, 20% yield) as a yellow oil; ^1^H NMR (400 MHz, CDCl_3_) δ 7.68–7.62 (m, 1H), 7.58–7.53 (m, 1H), 7.50–7.44 (m, 4H), 7.33–7.26 (m, 1H), 7.24–7.10 (m, 3H), 2.87–2.63 (m, 2H), 1.84–1.66 (m, 2H), 1.03 (t, *J* = 7.3 Hz, 3H); ^13^C NMR (101 MHz, CDCl_3_) δ 140.2, 136.2, 129.7, 129.4, 126.0, 125.2, 124.2, 122.4, 119.8, 119.5, 118.2, 110.6, 27.3, 23.4, 14.4; MS (ESI) *m*/*z*: 235.1[M]^+^.

##### 2-(1-phenyl-1*H*-indol-3-yl)ethanamine (**12c**)

General procedure III with (*E*)-3-(2-nitrovinyl)-1-phenyl-1*H*-indole (264 mg, 1.0 mmol), LiAlH_4_ (4.0 mmol, 152 mg) afforded 2-(1-phenyl-1*H*-indol-3-yl)ethanamine (**12c**) (40 mg, 0.17 mmol, 17% yield) as a yellow oil; ^1^H NMR (500 MHz, CDCl_3_) δ 7.47–7.42 (m, 1H), 7.36–7.32 (m, 1H), 7.27 (t, *J* = 1.7 Hz, 2H), 7.26 (s, 1H), 7.11–7.08 (m, 1H), 7.02–6.99 (m, 1H), 6.98–6.93 (m, 3H), 2.79 (dt, *J* = 60.2, 6.8 Hz, 4H); ^13^C NMR (126 MHz, CDCl_3_) δ 167.8, 139.9, 136.2, 129.7, 129.7, 126.2, 125.9, 124.2, 122.6, 120.0, 119.4, 110.7, 42.5, 29.6; MS (ESI) *m*/*z*: 236.1[M]^+^.

### 3.6. Docking Simulation

To predict a bind pose of DPIE derivatives in complex with human IL-1R, we performed a docking simulation using Vinardo [19] and improved the scoring function of AutoDock Vina for ranking and scoring. To prepare the target receptor IL-1R1, the X-ray structure of the human IL-1β- IL-1R1 complex was available from the protein databank (PDB ID: 4GAF) [20]. The IL-1β and water molecules were deleted, and deficient hydrogen atoms were added. The three DPIE derivatives were selected as **4d** of Type 1, **5a** of Type 2-1, and **9a** of Type 2-2 and were made by MarvinSketch [21]. While the derivative 9a was neutral in pH 7.4, 4d and 5a were totally protonated because the predicted pK_a_ values of **4d** and **5a** were 9.68 and 9.54, respectively. In our previous virtual screening study [18], two binding sites were assigned, and each interesting residue was Pro26 of Site 1 (S1) and Tyr127 of Site 2 (S2). After defining the appropriate search space for S1 and S2, a Vinardo docking simulation was carried out with IL-1R1 and three derivatives. Depending on the docking score, the best binding pose of each ligand was retrieved for further analysis. The interaction between IL-1R1 and docked compounds were analyzed using Chimera [22] with the utility of the analyzed complex module. 

## 4. Conclusions

In summary, we synthesized three types of DPIE derivatives, namely Types 1, 2, and 3. The DPIE derivative with a different N-aryl substituent was Type 1; the DPIE derivative with the alkyl or amino groups at the 3-position of indole core was Type 2; and the DPIE derivative without a phenyl ring at the 2-position of indole core was Type 3. All DPIE derivatives were successfully synthesized in good yields and fully characterized. Type 1 DPIE derivatives enhanced IL-6 expression relative to DPIE. The IL-8 expression level owing to Type 1 DPIE derivatives was similar to that produced by DPIE. These like mediators, which induced inflammatory cytokines, are important for host cell defense against pathogenic stimulation. Small molecules have positive features that activate IL-1β signaling. For example, the host defense peptide LL-37 enhanced the IL-1β-induced production of IL-6, IL-10, MCP-1, and MCP-3 in human peripheral blood mononuclear cells. These secreted proteins enhanced innate immune responses [23]. In addition, IL-1β activation was correlated with antimicrobial activity in macrophages [24]. 

Types 2 and 3 DPIE derivatives did not enhance the pro-inflammatory cytokine expression to the same extent as DPIE. Type 2 DPIE derivative displayed similar levels of IL-8 and IL-6 expression relative to IL-1β treatment alone. Compounds **9d**, **9e**, and **9f** slightly upregulated IL-6 expression relative to the IL-1β control; however, the IL-6 expression level was lower than that induced by DPIE. Most of the Type 3 DPIE derivatives did not enhance cytokine expression, but induced a slight decrease in IL-6 expression level relative to the IL-1β-treated control. As the inhibition of IL-1β signaling is important for the treatment of chronic inflammatory diseases, such as osteoarthritis and rheumatoid arthritis [25,26], the Type 3 DPIE derivative could be a good framework for anti-inflammatory agents. According to the docking study, whether the DPIE derivatives were selectively bound to any binding sites could enhance or reduce the binding interaction between IL-1R1 and IL-1β. Based on these results, three important structural conclusions were discovered for the activity of DPIE derivatives toward the receptor: (1) as the size of the N-aryl substituent increased, the binding affinity between DPIE derivatives and receptors increased; (2) the phenyl ring at the 2-position of indole was needed to maintain the strong affinity of DPIE toward the receptors; and (3) functionality at the 3-position of indole had no effect.

## Data Availability

Data is contained within the article and Appendix A.

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
