# Peer review of "Synthesis of DPIE [2-(1,2-Diphenyl-1H-indol-3-yl)ethanamine] Derivatives and Their Regulatory Effects on Pro-Inflammatory Cytokine Production in IL-1β-Stimulated Primary Human Oral Cells"

_molecules, 2022, doi:10.3390/molecules27030899_

Round 1

Reviewer 1 Report

The manuscript entitled “Synthesis of DPIE [2-(1,2-diphenyl-1H-indol-3-yl)ethanamine] derivatives and their regulatory effects on pro-inflammatory cytokine production in IL-1β-stimulated primary human oral cells” is dedicated to the study of DPIE derivatives as potential compounds affected different aspects of cytokine production.

In order to improve the manuscript, the following suggestions should be taken into account by the authors:

  1. The abbreviation DPIE should be disclosed in the abstract.
  2. In section Materials and Methods, please, add the subsection General in order to describe the whole equipment used to identify the structure of synthesized compounds. Provide this subsection with title of NMR and mass spectrometer along with their characteristics and producers.
  3. In all diagrams, please, delete the background lines.
  4. Authors should add Statistics subsection into Materials and Methods section.   

After this minor revision, I highly recommend the present manuscript for publication.  

Author Response

Dear Reviewer 2

We would like to express our sincere appreciation for the reviewers’ comments regarding this paper. Our point-by-point responses to each of the comments of the revision request are as follows.

Response to Referee 2’s comments and suggestions

Referee: 2

Reviewer 2’ Comment

1.The abbreviation DPIE should be disclosed in the abstract.

-> Response 1: As suggested, the full name of DIPE [2-(1,2-diphenyl-1H-indol-3-yl)ethanamine] was added in the abstract

2.In section Materials and Methods, please, add the subsection General in order to describe the whole equipment used to identify the structure of synthesized compounds. Provide this subsection with title of NMR and mass spectrometer along with their characteristics and producers.

-> Response 2: As suggested, the followings are added under the subsection of 3.4. Synthetic Method

General Information

All reagents were purchased and used without further purification. 1H spectra were recorded in CDCl3 or DMSO-d6 on 500 MHz NMR spectrometers and data are reported as follows: chemical shift, multiplicity (s = singlet, d = doublet, t = triplet, m = multiplet), coupling constants (Hz) and integration. 13C spectra were recorded in CDCl3 on 126 MHz NMR spectrometers and resonances (δ) are given in ppm. High resolution mass spectra was recorded on a time of flight (TOF) mass spectrometer. 1HNMR and 13CNMR were recorded on Agilent Technologics (U. S. A.) NMR spectrometer (500 MHz for 1H).

  1. In all diagrams, please, delete the background lines.

-> Response 3: As suggested all diagrams has clear background.

  1. Authors should add Statistics subsection into Materials and Methods section.   

-> Response 4: the statistical analysis was added into Material and methods section. And the significance was written in figure legend and Result section.

Many thanks again in advance for your further consideration of our manuscript.

Sincerely yours,

Sunwoo Lee

Reviewer 2 Report

In this submitted manuscript titled “Synthesis of DPIE [2-(1,2-diphenyl-1H-indol-3-yl)ethanamine] derivatives and their regulatory effects on pro-inflammatory cytokine production in IL-1β-stimulated primary human oral cells”,

Lim et al presented here a structure-activity relationship study of a previously identified molecule DPIE, which was found to synergistically increase inflammatory molecules and cytokine production (IL-6, IL-8, and COX-2) in IL-1β-stimulated GFs. In this manuscript, the authors synthesized three types of DPIE derivatives by introducing various substituents at the 1, 2, and 3 positions of the indole group in DPIE, and evaluated their effect in cytokine production  

 following IL-1β induction in gingival fibroblasts (GFs). As a result, the authors found that 2-phenyl group is essential and derivatives having larger functional groups than phenyl at 1-position seemingly displayed better synergistic pro-inflammatory cytokine production than DPIE.  

Overall, I find this submitted manuscript is very intriguing from a biological perspective, also synthetic chemistry in this submitted manuscript is quite solid, synthesized compounds are well characterized. I would recommend its publication in Molecules if the authors can kindly address my following comments:

Major comments:

1: In the quantification assay of mRNA levels of IL-6 and IL-8, “the cells were pre-incubated with DPIE derivatives and stimulated with 0.5 ng/mL human IL-1β _(Peprotech, Korea) for 8 h. The cells were then washed with phosphate buffered saline and lysed using TRIzol reagent (Invitrogen, USA) for mRNA purification.” Did the authors test the cytotoxicities of synthesized DPIE derivatives in advance or observe any toxicities of these derivatives in this assay? Since the cells were washed with PBS, only viable cells will be quantified, thus the assay result may be affected by cell death. I think the authors should provide such evidence by either showing no cell death or giving a western blot of both IL-6 and IL-8 in the presence of normalized GAPDH for some of the key compounds in each type of derivatives (especially for those compounds without synergy or lowering the cytokine expression).

Minor points:

1: In paragraph 2of the Introduction, the authors said “Despite their high ligand specificity, all are protein drugs including therapeutic antibodies.”, an exception may be caspase 1 inhibitors, for example, VX-765 as the authors mentioned earlier.

2: In paragraph 3 of the Introduction, “In our previous study, 60 molecules were evaluated to determine the effect of pro-inflammatory cytokine production in IL-1β-stimulated gingival fibroblasts (GFs) using in silico computational analysis [17].”, the reference should be 18, not 17.

3: Please correct NaB(OAc)3 in both Schemes and synthetic procedures, this should be NaB(OAc)3

4: In paragraph 1 of 2.2 (Effect of DPIE derivatives on Pro-Inflammatory Cytokine Production in IL-1β-Stimulated GFs), “To determine the activity of small molecules, GFs were stimulated with IL-1β for 1 h. Thereafter, the derivatives were administered at the indicated concentration. Cells were harvested and the mRNA expression of pro-inflammatory cytokines (IL- 6 and IL-8) was determined.”, this is not the same as the method as describe in 3.2 (Cell culture and reagents), please correct.

5: In paragraph 2 of 2.2 (Effect of DPIE derivatives on Pro-Inflammatory Cytokine Production in IL-1β-Stimulated GFs), “As shown in Figure 2, type 1 derivatives had better activity in IL-6 production than DPIE.”, I think the authors should be careful with this statement as 4b seemly not show biological significance, I suggest the authors add biostatistics and correct this statement if necessary.

6: In paragraph 3 of 2.2 (Effect of DPIE derivatives on Pro-Inflammatory Cytokine Production in IL-1β-Stimulated GFs), “Type 2-1 compounds were expected to exhibit hydrophilic and hydrogen bonding affinity between amino substituents and target protein, while type 2-2 compounds were expected to exhibit a hydrophobic interaction between the alkyl group substituents and target protein.”, I suggest the authors briefly describe the property of the binding pocket as well as the key amino acid residue that the 3-positioned aminoalkyl or linear alkyl chain may interact with.

7: If possible, can the authors please add synthetic yield for each step in each scheme?

Author Response

Dear Reviewer 1

We would like to express our sincere appreciation for the reviewers’ comments regarding this paper. Our point-by-point responses to each of the comments of the revision request are as follows.

Response to Referee 1’s comments and suggestions

Reviewer 1’s Comment

Major comments:

1: In the quantification assay of mRNA levels of IL-6 and IL-8, “the cells were pre-incubated with DPIE derivatives and stimulated with 0.5 ng/mL human IL-1β _(Peprotech, Korea) for 8 h. The cells were then washed with phosphate buffered saline and lysed using TRIzol reagent (Invitrogen, USA) for mRNA purification.” Did the authors test the cytotoxicities of synthesized DPIE derivatives in advance or observe any toxicities of these derivatives in this assay? Since the cells were washed with PBS, only viable cells will be quantified, thus the assay result may be affected by cell death. I think the authors should provide such evidence by either showing no cell death or giving a western blot of both IL-6 and IL-8 in the presence of normalized GAPDH for some of the key compounds in each type of derivatives (especially for those compounds without synergy or lowering the cytokine expression).

-> Response 1: As suggested, the cell viability assay was conducted. All results were added in the Supplementary Information

 Cell viability assay

Human GFs viability after the DPIE derivative treatment was investigated using the EZ-Cytox Cell viability assay kit (water-soluble tetrazolium salt method). The experiment was performed as manufacturer’s protocol. Briefly, the derivatives were treated for 24 hours after seeding the cells (1×104 cells per well). Then WST reagent solution (10 μl) was added to each well of a 96-well microplate that contained 100 μl of cells in the culture medium. The plate was then incubated for 1 h at 37 °C. The absorbance was measured at 450 nm using a microplate reader. At the same time, culture medium without cells were incubated for 1day to obtain the background signal. We calculated the final value: total signal−background signal=original signal, (original signal/control signal)×100=Survival (%).

Figure S1 The viability test was performed using water-soluble tetrazolium salt (WST) method on human GFs. The cells were incubated with DPIE derivatives for 24 hours were at indicated concentration. Group 4 chemicals (A) were tested at 2-8 μM, DPIE (B) was tested at 1 - 8 μM, and group 5(A), 9(B), 12(B) chemicals were tested at 25 - 75 μM. Control indicates hGFs treated with solvent, DMSO. Error bars means standard deviations from duplicated experiment.

Minor points:

1: In paragraph 2of the Introduction, the authors said “Despite their high ligand specificity, all are protein drugs including therapeutic antibodies.”, an exception may be caspase 1 inhibitors, for example, VX-765 as the authors mentioned earlier.

-> Response 1: we corrected the sentence to ‘most drugs are protein including therapeutic antibodies’.

2: In paragraph 3 of the Introduction, “In our previous study, 60 molecules were evaluated to determine the effect of pro-inflammatory cytokine production in IL-1β-stimulated gingival fibroblasts (GFs) using in silico computational analysis [17].”, the reference should be 18, not 17.

-> Response 2: As suggested, the reference number was corrected.

3: Please correct NaB(OAc)3 in both Schemes and synthetic procedures, this should be NaB(OAc)3H

-> Response 3: As noted, all “NaB(OAc)3” was corrected to “NaHB(OAc)3”

4: In paragraph 1 of 2.2 (Effect of DPIE derivatives on Pro-Inflammatory Cytokine Production in IL-1β-Stimulated GFs), “To determine the activity of small molecules, GFs were stimulated with IL-1β for 1 h. Thereafter, the derivatives were administered at the indicated concentration. Cells were harvested and the mRNA expression of pro-inflammatory cytokines (IL- 6 and IL-8) was determined.”, this is not the same as the method as describe in 3.2 (Cell culture and reagents), please correct.

-> Response 4: we corrected the sentence to “To determine the activity of small molecules, GFs were pre-incubated with the derivatives at the indicated concentration. Then cells were stimulated with IL-1β for 8 hours”.

5: In paragraph 2 of 2.2 (Effect of DPIE derivatives on Pro-Inflammatory Cytokine Production in IL-1β-Stimulated GFs), “As shown in Figure 2, type 1 derivatives had better activity in IL-6 production than DPIE.”, I think the authors should be careful with this statement as 4b seemly not show biological significance, I suggest the authors add biostatistics and correct this statement if necessary.

-> Response 5: we corrected the sentence to “4c and 4d in type 1 derivatives had better activity in IL-6 production than DPIE”.

6: In paragraph 3 of 2.2 (Effect of DPIE derivatives on Pro-Inflammatory Cytokine Production in IL-1β-Stimulated GFs), “Type 2-1 compounds were expected to exhibit hydrophilic and hydrogen bonding affinity between amino substituents and target protein, while type 2-2 compounds were expected to exhibit a hydrophobic interaction between the alkyl group substituents and target protein.”, I suggest the authors briefly describe the property of the binding pocket as well as the key amino acid residue that the 3-positioned aminoalkyl or linear alkyl chain may interact with.

-> Response 6:

We performed a docking simulation and the result was discussed in result section 3.3.

Actually, the amino group of Type 2-1 formed a hydrogen bond with Glu129 of IL-1R1 and the alkyl group of Type 2-2 was jammed between Pro26 and Phe130 of IL-1R1. The binding affinity of 5a (type 2-1) and 9a (type 2-2) was slightly different about 0.3 kcal/mol. Although the binding poses between 5a and 9a were different, the predicted score value seem similar and it is consistent with the experimental result, which was reduced for cytokine production.

The followings are added in the main text.

2.3. Molecular Docking study of DPIE derivatives and human IL-1R

    To predict a binding interaction of DPIE derivatives such as 4d of type 1, 5a of type 2-1 and 9a of type 2-2 with human IL-1R1, the docking study was performed using Vinardo software [19]. According to our previous study [18], the significant residue of the binding sites in IL-1R1 were defined by Pro26 of Site 1 (S1) and Tyr127 of Site 2 (S2). The region of S1 and S2 was located at the binding interface of IL-1R1 and IL-1β. Regarding the binding pocket S1, it was consisted in the inside hydrophobic hole and the cave was restricted. While the region of S2 coincided with the weak interface in the binding of the IL-1R1 and IL-1β and made up with the hydrophobic cores. After running docking calculations for two binding sites, the best pose was selected based on their docking score and analyzed concerning the derailed information about the binding mode in the cavity. Depending on the score value, the best poses of the derivatives 5a and 9a such as type 2-1 and type 2-2 were occupied in S1, while 4d of type 1 was bound to S2 (Figure 5). The binding affinity was -6.9 kcal/mol, -7.2 kcal/mol and -9.3 kcal/mol for 4a, 5a, and 9a, respectively. Regarding the binding pose of two different type 2 compounds in S1, 2-phenyl-N-phenyl indole moiety of 5a and 9s was surrounded by the hydrophobic sidechains of IL-1R1 such as Ile13, Leu15, Pro26, Pro28, Tyr127, and Phe130 in common. The difference of the binding pose between 5a and 9a was that the amine group of 5a (type 2-1) slightly formed a hydrogen bond with the charged sidechain of Glu129 and the alkyl group of 9a was lain between Glu129 and Phe130. Further, since two derivatives were deeply buried in S1 of IL-1R1, it could not affect the IL-1R1 and IL-1β interaction directly. Compared with 5a and 9a, interestingly the best pose of 4d was tightly stuck to S2. The indole group of 4d lay on the hydrophobic core including Pro26, Tyr127, and Phe130 and the 9,9-dimethylfluoren moiety was reached on Ile13 and Leu15. In addition, the ethanamine group of 4d formed a hydrogen bond with the backbone of Lys112 and the 2-phenyl group was stacked with the alkyl group of Lys112 in parallel. It is an important binding region formed of the hydrophobic cores and the derivatives 4d could construct in the stable complex with IL-1R1 because S2 seems like a crack between IL-1R1 and IL-1β. Noticeably, the possible binding poses of 4d were only occurred in S2 and totally the binding affinity was lower than that of 5a and 9a. Therefore, the analogue expansion of 4d could be promised to enhance the protein-protein interaction of IL-1R1 and IL-1β.

Figure 5. The predicted binding pose of DPIE derivatives such as 4d (type 1, purple), 5a (type 2-1, magenta), and 9a (type 2-2, yellow) in complex with IL-1R1. The ligand binding sites, S1 and S2 represented by a red dash line were located at the interface of IL-1R1 (rainbow) and IL-1β (pink) interaction. The derivatives 5a and 9a was well suitable in the binding site S1 and surrounded with the hydrophobic sidechains such as Ile13, Leu15, Pro26, Pro28, Tyr127, and Phe130 in common. Compared with the binding mode of 5a and 9a, the amino group of 5a formed a hydrogen bond with the charged sidechain of Glu129 and the binding affinity of 5a was more stable than that of 9a. the derivatives 4d was tightly stuck to S2 of the receptor IL-1R1 and the charged amine group of 4d made a hydrogen bond with the backbone of Lys112 of IL-1R1. The gray dash line represents a hydrogen bond.  

7: If possible, can the authors please add synthetic yield for each step in each scheme?

-> Response 7: As suggested, synthetic yields were added in the reaction scheme.

Many thanks again in advance for your further consideration of our manuscript.

Sincerely yours,

Sunwoo Lee

Round 2

Reviewer 2 Report

In this submitted revision, Lim et al have presented a decent response to my comments raised in the original submission and successfully addressed all my concerns. I would recommend its publication in Molecules if the authors can kindly address my following minor comments:

Minor points:

1: I suggest the authors add their new toxicity result to either the main text or supplementary information, and please briefly state that no apparent toxicity was observed for all the compounds at tested concentrations.

2: Please add the company information for HR-MS in the general information; silica gel information for flash chromatography; TLC information for monitoring the reaction.

3: Please add HPLC purity information to all the biologically tested compounds.

Author Response

Dear Reviewer 2

We would like to express our sincere appreciation for the reviewers’ comments regarding this paper. Our point-by-point responses to each of the comments of the revision request are as follows.

Response to Referee 2’s comments and suggestions

Referee: 2

Reviewer 2’ Comment

1: I suggest the authors add their new toxicity result to either the main text or supplementary information, and please briefly state that no apparent toxicity was observed for all the compounds at tested concentrations.

-> Response 1: As suggested, the following was added in the main text.

“The concentration of small molecules were determined after cell cytotoxic assay (Figure S1), the molecules were applied with non-cytotoxic concentration.”

2: Please add the company information for HR-MS in the general information; silica gel information for flash chromatography; TLC information for monitoring the reaction.

-> Response 2: As suggested, the followings are added in the general information.

“Analytical thin layer chromatography (TLC) was performed on silica gel (Merck Kieselgel 60 F254) pre-coated aluminum plates and the products were visualized by short-wave (254 nm) or long-wave (360 nm) UV light. Flash column chromatography was performed using silica gel (Merck, Kieselgel 60, 230-240 mesh).”

The information of HRMS was added: “(JEOL, Model: JMS-T200GC).“

3: Please add HPLC purity information to all the biologically tested compounds.

-> Response 3: As suggested, the following is added in the general information.

“The purity of all DPIE derivatives was determined by HPLC (Waters ACQUITY (UPLC), SQD2) and confirmed to 100%.”

Many thanks again in advance for your further consideration of our manuscript.

Sincerely yours,

Sunwoo Lee
